# Automatic Calibration of an Industrial RGB-D Camera Network Using Retroreflective Fiducial Markers

**DOI:** 10.3390/s19071561

**Published:** 2019-03-31

**Authors:** Atle Aalerud, Joacim Dybedal, Geir Hovland

**Affiliations:** Department of Engineering Sciences, University of Agder, 4879 Grimstad, Norway; joacim.dybedal@uia.no (J.D.); geir.hovland@uia.no (G.H.)

**Keywords:** 3D sensors, time-of-flight, automatic calibration, retroreflective markers, ambiguity problem

## Abstract

This paper describes a non-invasive, automatic, and robust method for calibrating a scalable RGB-D sensor network based on retroreflective ArUco markers and the iterative closest point (ICP) scheme. We demonstrate the system by calibrating a sensor network comprised of six sensor nodes positioned in a relatively large industrial robot cell with an approximate size of 10 m×10 m×4
m. Here, the automatic calibration achieved an average Euclidean error of 3 cm at distances up to 9.45
m. To achieve robustness, we apply several innovative techniques: Firstly, we mitigate the ambiguity problem that occurs when detecting a marker at long range or low resolution by comparing the camera projection with depth data. Secondly, we use retroreflective fiducial markers in the RGB-D calibration for improved accuracy and detectability. Finally, the repeating ICP refinement uses an exact region of interest such that we employ the precise depth measurements of the retroreflective surfaces only. The complete calibration software and a recorded dataset are publically available and open source.

## 1. Introduction

With the increasing quality and availability of 3D sensors, there is a growing interest in using such sensors in industrial applications. The use of these sensors could open up new and more flexible applications, for example within human-robot interaction (HRI), in addition to the traditional applications with safety fences and restricted access for humans. Examples may be robotic cells in a factory environment or outdoor facilities such as the drill floor of an offshore drilling rig. While humans may be physically isolated from robots in a factory environment, this is often not the case in offshore drilling where human workers collaborate with heavy robots supervised by an operator. As this operator is unable to monitor the entire hazard zone, there is a need for a 3D monitoring system to detect personnel or other foreign obstacles. Depending on the type of collaboration, robots may be programmed to re-plan, slow down, or stop when humans are present in their environment. If the application requires physical contact or close collaboration, such as in an assembly line, power and force limiting may be needed. In industrial environments requiring simultaneous tasks from a human and a robot, but not close collaboration, speed and separation monitoring can be applied.

One of the first steps required when setting up an industrial cell instrumented with 3D sensors is calibration, i.e., to match the internal coordinate systems for each sensor such that the point cloud from each sensor can simply be added together to form a global point cloud for the entire industrial cell. In rough industrial environments such as offshore installations, external factors may cause the sensors to become misaligned. Although the environment and sensor mounting positions may be known and valid, a refinement of the orientation may be needed at given intervals or when misalignment is detected. Such an automatic calibration scheme should not rely on personnel in any way as the purpose is to increase the autonomy of the industrial installation. It should be possible to perform calibration without stopping the operation of the facility. Thus, no robots or moving machines should be used to position calibration targets as this could interfere with ongoing operations. Calibration accuracy should be sufficient to position a human within 10 cm, such that safety applications can be made based on a speed and separation monitoring scheme.

Obtaining the camera pose from images requires finding the correspondences between known points in the environment and their camera projections [1]. Such key points [2,3] or calibration objects [4,5,6] may be found in the image naturally, or they may be inserted artificially [7,8,9,10,11,12,13,14,15]. Even if natural landmarks are the preferred choice, as they require no intervention in the environment, artificial landmarks have some clear benefits such as higher precision, robustness, and speed [16].

Fiducial markers are artificial landmarks added to a scene to facilitate locating point correspondences between images, or between images and a known model [17]. Such markers can for example be spherical [7,8,9,10], circular [12], simply reflective surfaces [18], or square [19]. As the circular systems can provide only a single point with a high degree of accuracy, multiple circular markers are needed for full pose estimation [17]. Square fiducial markers, on the other hand, provide a key point for each of the four corners which are sufficient to perform the camera pose estimation [1,16]. Squared-based fiducial markers typically have an external black border for corner detection and an internal pattern used for identification, for example ARToolKit [20], ARTags [19], ArUco [1,21,22] and AprilTag [23,24]. Although these markers have a similar function, their performances differs in terms of computing time and detection rate. For details regarding the marker performance, the reader is referred to the respective papers of the markers, [1,19,20,21,22,23,24]. Currently, the most popular marker in academic literature is probably the ArUco [1] as this is included in the popular open source computer vision library OpenCV [25].

Once the key points are found, camera systems typically need to solve the inverse problem known as the perspective from N points problem (PnP). According to [26], the “gold-standard” solution to the PnP consists in estimating the six parameters of the transformation by minimizing the norm of the reprojection error using a non-linear minimization approach such as a Gauss–Newton or a Levenberg–Marquardt technique. However, as the corner positions of squared-based markers are detected using rectangles and lines, the accuracy of their sub-pixel corner positions are not as accurate as the center point of a circular marker. In theory, the pose of a camera w.r.t. four non-linear and co-planar points can be uniquely determined. However, in practice, there is a rotation ambiguity that corresponds to an unknown reflection of the plane about the camera’s *z*-axis [27]. That is, a marker could project at the same pixels on two different camera locations. In general, the ambiguity can be solved, if the camera is near to the marker. However, as the marker becomes small, the errors in the corner estimation grow, and ambiguity comes as a problem. For 3D cameras, such as RGB-D, an additional refinement is typically performed utilizing the depth information [6,8,9,10,11,28].

## 2. Related Work

Jin et al. [29] mitigate the ambiguity problem by fusing the depth and RGB information of an RGB-D camera to increase the robustness of the pose estimation when detecting an AprilTag. The proposed method has three distinct components. First, a plane is estimated based on the rectangular patch of points in the depth image bounded by the approximated corner pixels of the marker. Secondly, an initial pose is estimated by minimizing the least squares error between a projection of the corner points and the corner points of the plane. Lastly, the pose is refined by minimizing the reprojection error using a constrained optimizing function. Here, the constrained region is defined by the uncertainty in the initial estimate which is characterized by the covariance of the plane parameters. The method is demonstrated using simulations and an experiment with a 7 cm AprilTag at a distance of 0.65
m away from the sensor. The results show that the method can reduce the error due to ambiguity significantly. However, Jin et al. assume that the depth and RGB cameras are registered to the same frame. Relying on a perfect cross-calibration between RGB and depth may introduce errors at long distances.

Squared-based fiducial markers may fail when the image is blurred due to camera movements, under non-uniform light, or when markers are seen at low resolution. Mondéjar-Guerra et al. [16] propose a method that handles fiducial marker identification as a classification problem using machine learning to increase the performance under difficult image conditions. The tested image conditions are motion blur, camera defocus, camera overexposure, non-uniform lighting, and small scale markers. By comparing three machine learning approaches with ArUco and AprilTags, it was shown that the machine learning approaches surpass the other methods in marker identification. Further, Mondéjar-Guerra et al. observe that ArUco and AprilTags are highly restrictive during the marker identification process, leading to high precision at the expense of a low recall and that ArUco is orders of magnitude faster than the other methods. They also note that although the method can correctly detect and identify the marker, the estimation of the 3D location depends on the corner locations which are still affected by blurring.

A markerless automatic calibration of monocular cameras is proposed in [4]. Here, multiple detections of a person’s head and foot position are used to estimate vanishing points and lines. These points and lines are used to calculate the extrinsic camera parameters. This method could be used for automatic re-calibration. However, the use of pedestrians as natural landmarks is not desirable in the proposed application as this would require personnel to be present during calibration.

Another calibration method for monocular cameras is presented in [7]. Here, Guan et al. calibrate multiple cameras using a lighted sphere in a slightly darkened room. The sphere is placed in the overlapping volume at minimum three distinct non-collinear locations where each camera captures an image. One advantage of using a lighted sphere is that simple thresholding is sufficient to segment the sphere blobs from the background. The first camera’s coordinate system is used as the world coordinate system, and all other cameras are related to it using pairwise calibration. Guan et al. placed the cameras at the height of about 3 m in each corner of a room (8.6×4.8
m) and were able to achieve a multi-camera triangulation error of 4.5
cm.

Spheres can also be used for calibration of RGB-D sensor networks as their center point can be calculated regardless of view [8,9,10]. In [8] Ruan and Huber, present a method that places three spheres in precisely measured positions. Each sensor detects and segments the spheres before estimating their center positions. Further, extrinsic camera parameters are calculated using singular value decomposition (SVD). As the described method uses known positions, sensors are inherently calibrated w.r.t. a world coordinate system. However, the spheres clutter the work environment.

Su et al. [10] mitigate the problem of a cluttered environment as they use a moving sphere. The proposed method uses a large ball where time-synchronized sensors observe the center of this ball as a function of time. The calibration procedure is based on an optimization procedure using this common 3D path of the ball. The proposed approach works well when the sensor network is calibrated once, and a human operator can be sent inside the sensed volume to move the ball. However, in an industrial setting where automatic re-calibration of the sensor network is desired at regular time intervals, or when it is desirable to detect when a sensor node has moved, the method proposed in [10] may not be feasible.

A system for calibrating 28 Asus Xtion 3D sensors based on structured light and six Hokuyo 2D lasers is presented in [5]. This method uses the tracking of pedestrian heads for sensor calibration. As 3D sensors based on structured light cannot have a high degree of overlap due to the interference of the emitted structured light between the sensor nodes, they implemented the 2D laser sensors to increase the number of shared observations.

In [30] a solution is presented where vibration motors were attached to each sensor to mitigate the interference problem. However, this solution reduces the accuracy of the sensors as well as increases complexity. In our work, 3D sensors based on structured light were avoided, and sensors based on the time-of-flight (ToF) principle were used instead, which allows for overlap between the different 3D sensors. Additionally, [31] states that the outdoor applicability for structured light based depth sensors is usually hard to achieve.

OpenPTrack [11] is an open source software for people detection and tracking, but it also includes calibration of RGB-D camera networks. The extrinsic calibration is two-fold using an initial and a refined calibration process. The initial calibration is performed automatically while moving a large checkerboard in front of the cameras. After all of the cameras are pairwise calibrated, the checkerboard can be put on the ground such that the ground plane w.r.t. each camera is calibrated. The second part refines the calibration utilizing the people tracking software. One single person moves around in the environment while all the sensor nodes record the calculated track. Then, the timestamp replaces the *z*-coordinate of the recorded track such that the entire track can be pairwise refined using iterative closest point (ICP). Although this calibration procedure seems to work quite well, it requires both a large checkerboard and multiple human actions. Hence, it should not be applied in environments without trained personnel.

Desai et al. [6] proposed a method using skeleton tracking. Here, the sensors identify the human skeleton joints and use the 3D joint positions as point correspondences between cameras on a pair by pair basis. Seven Microsoft Kinect for Xbox One (Kinect V2) sensors were used to capture an area of 3 m×4
m from a maximum height of 2 m. Using this setup, Desai et al. compared the performance of their method against three traditional methods using a checkerboard [13], the intersection of a plane [14], and block pattern and ICP [15]. The average Euclidean distance errors obtained by using the traditional methods were 8.45
cm, 4.90
cm, and 4.12
cm, respectively. Using the skeleton approach gave an average Euclidean distance error of 1.5
cm. The approach in [6] can re-calibrate automatically as long as a human is present in the scene. However, it does not include an automatic method to calibrate the camera network w.r.t. a world coordinate system. Furthermore, accurate joint tracking cannot be guaranteed beyond 3 m.

In the aerial robotics testbed called the flying machine arena (FMA) [32] use a motion capture system for advanced quadrocopter navigation and interaction such as ball juggling and cooperative ball throwing and catching. Attached to each vehicle is a set of at least three uniquely configured retroreflective markers observed by the motion capture system. Using this method, the FMA provides precise and robust tracking for multiple quadrocopters in a large volume. However, motion capture systems are costly and typically depend on markers placed directly on the objects to be tracked.

Hartmann et al. [33] present an optical tracking concept using a spatial array of retroreflective fiducial markers for vertical precision landing of an unmanned aerial vehicle (UAV). The described method illuminates and detects the markers using a light source mounted next to a monocular camera on the UAV. The retroreflective marker coating enabled the system to work in varying ambient light as well as in complete darkness. The median position error was 3 cm, 10 cm, and 21 cm from a distance of 1.0
m, 1.5
m, and 2.5
m respectively. For the presented application where the UAV may refine the position as it approaches, this accuracy is sufficient. However, in an industrial environment where the distance between the camera and the marker is large, a higher accuracy will be required.

Tong and Barfoot [18] also used retroreflective landmarks. Four 1.2×1.2
m large retroreflective signs, with no marker pattern, were mounted around the walls of a dome structure with a workspace diameter of 40 m such that the position of a rover equipped with a 3D scanning laser was calculated based on the distances to a minimum of three landmarks. The described method exploits the sensor’s ability to identify retroreflective objects in the scene due to their unusually high return intensities. The results show an RMS error of 2 cm to 6 cm in positioning and 0.2${}^{\circ }$–0.5${}^{\circ }$ in orientation.

Ujkani et al. [28] present a benchmark and accuracy analysis of 3D sensor calibration in a large industrial robot cell. Here, a rough calibration using a single ArUco marker is performed. Then, the calibration is refined using three markers, region of interest (ROI) and ICP. Finally, a second refinement is applied using the full point clouds. The calibration of sensors is performed pairwise, exploiting overlap in the point cloud data. For a volume measuring 10 m×15 m×5
m, a typical Euclidean distance error of 5 cm to 10 cm was achieved using six sensor nodes. Although the method presented in [28] worked well, it contains several drawbacks as follows: (1) the method requires some interaction by the operator, (2) there is no automatic camera to world calibration, (3) the solution could not easily refine the calibration when cameras observed the scene from different perspectives, (4) due to the various cross-calibrations involved, the process was susceptible to inaccurate RGB-to-depth calibration and (5) small and non-reflective ArUco markers were used.

## 3. Experimental Setup

The Industrial Robotics Lab (IRL) at the University of Agder depicted in Figure 1 is the industrial volume to be mapped by an automatically self-calibrating sensor network. This lab functions as a large robotic cell where both industrial robots and humans need to operate. Thus, it is crucial that the positions of the human workers are known so that safety systems can be developed for the robots. The specific setup of the sensor network and the lab have been described in [34] where a manual calibration method was used. However, key information is included here for completeness. The previously mapped volume was based on a floor surface of approximately 10 m×15
m and a sensor height of nearly 5 m. To increase data density, the sensors were moved such that an area of 10 m×10
m is mapped from a height of around 4.3
m.

The proposed environment was chosen to resemble many possible industrial applications such as the offshore drilling case described in Section 1. With such applications in mind, the sensor network was previously tested outdoors at a test facility used for drilling equipment. This facility, as depicted in Figure 2, has similar dimensions to the drill floor.

The sensor network comprised of six nodes in waterproofed cabinets. Each cabinet was equipped with a Kinect V2 RGB-D camera based on an active infrared (IR) sensor using the time-of-flight principle. The depth cameras had a specified operating range of 0.8 m to 4.2 m, an optical field of view of 70${}^{\circ }$(H) ×  60${}^{\circ }$(V) and an image resolution of 512 px×424 px [35]. Even if it is well beyond the specified range, the sensors were used with a range of up to nine meters (normal to the image plane) in work presented in this paper so that the entire required volume can be mapped. Note that the specified range of the Kinect V2 sensor applies to reliable human joint tracking only and that depth data is usable up to the required range [36]. Some of the nodes had additional sensors, but these were not considered here. Further, each node contained an NVIDIA Jetson TX2 Development board. These boards process the sensor data before sending requested data via Gigabit Ethernet to a personal computer (PC). The PC used in this paper was equipped with a 3.6
GHz Intel Core i7-7820x central processing unit (CPU), 32 GiB system memory and an NVIDIA GeForce GTX 1080 Ti graphics processing unit (GPU), which in turn has 3584 CUDA cores and 11 GiB memory. The communication between the sensor nodes and the central PC is handled by the open-source middleware Robot Operating System (ROS) [37] running on Ubuntu 16.04.

## 4. Calibration Method

In this paper, we propose a fully automatic and scalable calibration of an RGB-D sensor network that requires no human involvement. The method had similarities to previous work presented in [28] as ArUco markers, ICP, and the same sensor network and environment were used. However, we introduced several new aspects and overcome all the drawbacks described in Section 2. A comparison of the methods is given in Section 6.

There are currently some prerequisites for our method. First, we assumed that all cameras were intrinsically calibrated. We also assumed that all cameras face a common volume where fiducial markers in known positions are visible to all sensors.

### 4.1. Retroreflective Markers

Detection of ArUco markers relies on accurate corner detection. This task is simplified when using a high definition (HD) camera. However, in this work, only standard definition (SD) was used. The Kinect V2 has an HD color camera, but this was offset from the IR and depth camera. Thus, the depth map must be mapped to the color map or vice versa to match depth and color information. The accuracy of this alignment depends on the extrinsic calibration between the two internal cameras. As the depth map will be used in future applications, it was deemed important to avoid mapping this to a second camera frame which may introduce a source of error. Thus, the color image was mapped to the IR camera frame. The ROS package IAI Kinect2 [38] handled the sensor input in the sensor node and performs the described mapping. The intrinsic camera calibration and color to IR calibration was previously performed according to IAI Kinect2 documentation. As the color image was mapped to align with the depth map, the remaining SD resolution was only 512 × 424. This complicated an accurate corner detection of the ArUco marker.

The calibration targets to be used needed to be easily detectable, even when using low-resolution images. Thus, to improve detection, the markers were made with a cell size of 100 mm. Further, a marker dictionary should be chosen such that it enables a robust detection. Garrido-Jurado et al. explained in [21] that the most relevant aspects to consider during the design of a marker dictionary are the false positive rate, the false negative rate, and the inter-marker confusion rate. Further, they argued that a key aspect of such dictionaries is the inter-marker distance, which is the minimum Hamming distance between the binary codes of the markers, considering the four possible rotations. To avoid false detections or inter-marker confusion, the minimum inter-marker distance can be increased by selecting a marker with many internal bits or a small dictionary containing few IDs.

As a large cell size is required to get detections using SD images at a long distance, it is not an option to use markers with many bits as the markers would become very large. However, reducing the dictionary size is possible as many unique IDs are not needed. The chosen standard ArUco dictionary is DICT_4X4_50 which is a 4 bit×4 bit marker with 50 different IDs. This is more than sufficient for the intended application. Nevertheless, the choice of the marker dictionary should be evaluated for large-scale implementations.

We apply a 100 mm white border around the marker which yields a total board size of 800 mm×800
mm. A second way to improve detection is to ensure that the marker is planar. Ideally, a glass plate could be used for this, but this would be heavy and fragile. Thus, the boards were produced using a sandwich-structure of extruded polystyrene foam with a one-millimeter thick aluminum sheet on both sides. Accuracy and precision of the Kinect V2 have been thoroughly covered such as in [39]. Here, it was also documented that the reflectivity of an object influences the depth measurement greatly. More reflective colors yielded a lower standard deviation on the depth values. Such benchmarks were not performed in this work. However, the advantages of using retroreflection in a multi-sensor calibration scheme will be highlighted. To increase depth accuracy and IR visibility, we printed the ArUco patterns on 3M Engineer Grade Reflective Sheeting Series 3200 [40]. Further, the 3M sheeting was covered with a matte anti-glare film to reduce unwanted reflections and glare. How the calibration targets look in an IR image is shown in Figure 3.

### 4.2. Main Calibration Scheme

The main parts of the developed calibration program are shown in Algorithm 1. First, a world map was created by converting computer generated padded ArUco marker images to point clouds. The point clouds were transformed into known poses in the world coordinate system and combined as illustrated in Figure 4. An advantage of generating such a world map was that one does not need to consider the angle between sensors facing the same target.

As calibration was not performed pairwise, all sensors may be calibrated simultaneously. The calibration was initiated by recording the camera matrix, SD color images, depth maps, and point clouds. The data was processed such that all markers are detected. This information is used to crop the depth maps and calculate point clouds such that the resulting point clouds will have the same ROI. By using quaternion averaging as described in [41], an average pose is calculated based on the best marker pose estimations returned by a PnP algorithm. This is aligned with the averaged pose in the world map shown in Figure 4b. The function processRecordedData is further explained in Algorithm 2 where the best marker pose selection is detailed in Algorithm 3. To refine the calibration, the ROI cropped point clouds were transformed using the ArUco-based sensor pose described above. The point clouds are then registered with the world map using a repeating ICP where each repetition uses new refinement parameters. The function refineTransformation is further described in Algorithm 4.

**Algorithm 1** Pseudo code of the main calibration.
  1:**for** all markerId **do**  2:    worldCloud += simulateMarker(worldMarkerPoses[markerId], markerId)  3:avgWorldMarkerPose = getAverageTransformation(worldMarkerPoses)  4:**for** all sensor **do**  5:    sensorData[sensor] = recordRosTopics(sensor)  6:    markerPoses = processRecordedData(sensorData)  7:    avgMarkerPose = getAverageTransformation(markerPoses)  8:    sensorPose = avgWorldMarkerPose · avgMarkerPose${}^{-1}$             ▹ ArUco-based sensor pose  9:    sensorCloud = transformCloud(sensorData.croppedCloud, SensorPose)10:    sensorPoseRefined = refineTransformation(sensorCloud, worldCloud)    ▹ ICP refined pose


**Algorithm 2** Pseudo code of data processing function.
  1:**function**processRecordedData(sensorData)  2:    arucoParam = setArucoParam()  3:    **for** each colorImg in sensorData.colorImgs **do**  4:        [corners, id] = detectMarkers(arucoParam, Img)  5:        markerMap.insert(corners, id)  6:    evalDetectionRate(markerMap, sensorData.colorImgs)       ▹ Low detection rate yields a warning  7:    **for** each matching set of id in markerMap **do**  8:        avgCorners = calcAvg(markerMap.corners)  9:        mask[id] = makeCropMask(avgCorners)                                   ▹ Creates padded cropping mask10:        depthPoints3D = getMCP(avgCorners, sensorData.depthImg, sensorData.intrinsics)11:        singleMarkerPoses = calcPose(markerMap.corners, arucoParam, sensorData.intrinsics)12:        index = findBestPose(singleMarkerPoses, depthPoints3D, arucoParam)13:        markerPoses[id] = singleMarkerPoses[index]14:    clear(croppedCloud)15:    **for** N depthImg in sensorData.depthImgs **do**16:        croppedDepthImg = cropDepth(depthImg)17:        croppedCloud += calcSensorCloud(croppedDepthImg, sensorData.intrinsics)18:    sensorData.croppedCloud = croppedCloud19:    **return** markerPoses


**Algorithm 3** Pseudo code of the findBestPose function.
  1:**function**findBestPose(poses $T_{M}^{O}$, 3D corner points $P_{D}$)  2:    calculate transformation from marker origin to each corner, $T_{C}^{M}$  3:    **for** all poses(i) **do**  4:        **for** all 4 corners **do**  5:           $T_{C}^{O}=T_{M}^{O}\;\cdot \;T_{C}^{M}$  6:           $P_{T}=T_{C}^{O}\cdot translation$  7:           $D^{2}=D^{2}+{\left(P_{T}-P_{D}\right)}^{2}$  8:        **if**
$D^{2}$<score **then**  9:           score = $D^{2}$10:           index = i11:    **return** index


### 4.3. Mitigating the Ambiguity Problem

The proposed mitigation of the ambiguity problem is described in the function findBestPose shown in Algorithm 3, but first an explanation of the function processRecordedData where findBestPose is called for.

Initially, in Algorithm 2, ArUco parameters were initialized and the sensor data regarding one of the sensors is received. Then, these were used to detect ArUco markers in all color images using the ArUco library in OpenCV. For the calibration logs summarized in Table 1, 400 color images were used. All detected marker IDs and corners were then inserted into the markerMap. The number of detections was evaluated for each marker and compared with the number of images so that a warning was triggered if the detection rate was low for any of the markers. For every group of identical marker sets in markerMap, a set of actions were executed (e.g., markers in the group of markers with ID 40). The first action is to calculate the average corner pixels of the marker group. These are decimal values describing the *x* and *y* sub-pixel on the color image where the corner can be found. Then, a padded cropping mask is created to match the average position of the marker in the color image. (Note, this will not be used until later.) The same averaged corner pixels were used to extract the median corner point (getMCP). These 3D points are calculated in getMCP by extracting the depth value of the nearest pixel to the averaged corner pixel for every corner in every depth image. The median of the depth value is used to calculate the 3D point position using the camera intrinsic matrix. The function calcPose calculates the estimated sensor poses for each marker detection based on the detected corners, ArUco parameters and camera intrinsic matrix. The pose estimations were subject to the ambiguity problem. Hence, a function was made to find the best of the proposed transformations.

**Algorithm 4** Pseudo code of the refineTransformation function.
  1:**function**refineTransformation(sensorCloud, worldCloud)  2:    dScore = 1000  3:    score2 = 1000  4:    r = 0                                                                                                                                           ▹ round number  5:    $T_{W}^{S}$ = Identity                                                                                     ▹ Transformation from sensor to world  6:    cloud1 = sensorCloud  7:    **while** dScore>$ICP_{CC}$ and r<$ICP_{R}$
**do**  8:        r++  9:        [$T_{2}^{1}$, conv, score1] = alignClouds(Cloud, worldCloud, ($ICP_{D}$/r), ($ICP_{IT}$·r), $ICP_{TE}$)10:        **if** conv **then**11:           $T_{W}^{S}$ = $T_{2}^{1}\;\;\cdot \;\;T_{W}^{S}$12:           cloud1 = transformPointCloud(cloud2, $T_{2}^{1}$)13:           cloud1 = cloud214:           dScore = abs(score1-score2)/score115:           score2 = score116:        **else**17:           Error message18:    transformedSensorCloud = transformPointCloud(sensorCloud, $T_{W}^{S}$)19:    **return**
$T_{W}^{S}$


The squared Euclidean distance between a depth based corner point, $P_{D}$, and a projected corner point, $P_{T}$ is calculated in findBestPose, as shown in Algorithm 3. Here, the projected corner point was based on the estimated transformation, $T_{M}^{O}$. Further, as $T_{M}^{O}$ is the translation from the camera to the marker origin, a transform to the corner $T_{C}^{M}$ was applied. The squared Euclidean distances for all four corner pairs are summed as a score and the index of the transformation that yields the lowest score is returned.

In Algorithm 2, the index was used to save the best ArUco-based marker pose. Then, for a selection of the depth images, the cropping masks were applied and ROI point clouds were calculated and combined. Using a padded masked crop instead of e.g., a rectangular crop, as illustrated in Figure 5, ensures that only points from the planar common surface are used for point cloud calculation.

The point clouds were, at this point, accurately cropped so that all points had been captured from a retroreflective surface. Further, as described in Algorithm 1, they were transformed using the ArUco-based sensor pose which has been calculated based on all three markers. Typically, if a pose estimation was used from a single marker, and this estimation had a rotation error due to the ambiguity problem, a calibration refinement based on ICP may fail due to the lack of corresponding points. However, by using an average of multiple markers, one orientation may fail without failing the calibration refinement.

### 4.4. Repeating the ICP Refinement

The calibration refinement described in Algorithm 4 was built around the ICP algorithm from the Point Cloud Library (PCL) [42]. The ICP algorithm calculates a rigid transformation between two point clouds by minimizing the sum of Euclidean distances between the two. In this case, one was the cropped sensor point cloud, and the other was a generated world point cloud. The ICP transformation is estimated based on SVD and has several termination criteria in general: (1) number of iterations has reached the user imposed limit of iterations ($ICP_{IT}$), (2) the epsilon (difference) between the previous transformation and the currently estimated transformation is smaller than a user imposed value ($ICP_{TE}$) and (3) the sum of Euclidean squared errors is smaller than a user-defined threshold ($ICP_{score}$).

In this paper, $ICP_{score}$ was not used, as this functionality was solved in Algorithm 4 using a new user defined threshold, $ICP_{CC}$. The refineTransformation function received the ROI cropped, ArUco-transformed, sensor point cloud, sensorCloud , and the world map point cloud, worldCloud. The transformation between them, $T_{W}^{S}$, was initially set to identity and the sensor cloud was copied to a temporary point cloud. Then, while two configurable criteria were true, a procedure was repeated. The first criterion was that the absolute change in ICP score must be larger than the threshold $ICP_{CC}$. The second criterion ensured that the total number of rounds did not exceed $ICP_{R}$, which was a user-defined maximum number of rounds to perform. The round number, *r*, was incremented and used such that every ICP execution used new parameters. The first ICP execution in *alignClouds*, the parameters $ICP_{D}$, $ICP_{IT}$, $ICP_{TE}$ are used directly, as *r* increases, the search radius, $ICP_{D}$, is reduced and the amount of allowed iterations, $ICP_{IT}$, is increased. This method ensures an increasing refinement for each round. If the algorithm converges, the transformation, $T_{W}^{S}$, was updated by multiplication with the refinement, $T_{2}^{1}$. By applying this refinement,cloud1 was transformed to cloud2 and moved back into cloud1. In this way, cloud1 matched the worldCloud better with each round. Finally, a transformed sensorCloud was saved and the final transformation, $T_{W}^{S}$, was returned.

### 4.5. Validation

Three calibration logs were created using slightly different parameters for the calibration program. These parameters are summarized in Table 1. To validate the calibration results from the log files, twelve ground truth targets were positioned using a Leica Absolute Tracker AT960. These retroreflective validation markers were made of the same 3M sheeting as the ArUco markers. Although the Leica AT960 yielded sub-millimeter accuracy, one-millimeter accuracy was assumed for the process of positioning the markers. Two different validation marker configurations are shown in Figure 6.

To calculate where each sensor estimates the validation markers to be, the correct pixel in the depth map needed to be selected for depth extraction. In the same manner as when calculating corner points in Section 4.3, the correct pixel must first be found in the color image as this is easier to recognize. Thus, the pixel at the center of each validation marker was selected, and the 3D coordinate was calculated using the sensor intrinsic matrix and the refined calibration transformation. However, in some cases there was no obvious single pixel at the center of the marker as shown in Figure 6c. In such cases, two pixels were used to calculate an averaged position to avoid artificially high errors due to low pixel density.

As an additional validation of the sensor poses found by the automatic calibration, we measured the sensor positions. Although we used the Leica AT960, we have identified several deficiencies in our method. Firstly, the sensors were mounted inside waterproof cabinets, and we assumed a constant offset from the measured position on the cabinets to the actual sensor frame. In reality, there may have been some small variations in the mounting position of the sensor in the cabinets. This constant offset was applied using the angles found by the automatic calibration as we were not able to measure the angles of the cabinet. Further, we have mounted the sensor cabinets on vertical rails attached to horizontal cable ladders for flexible positioning. As the positions of the sensors were more than 4 m above the floor, it was necessary to lean a ladder against this structure. It was thereby causing the cable ladder to deflect slightly such that the position of the cabinet was offset during measurement. We do not know the total offset during measurement, but it appeared to be less than 5 cm. Recognizing these deficiencies, we include the measured positions in Section 5, but we encourage the reader to treat these measurements as indicative.

## 5. Experimental Results

The complete software and log-files are archived and available on Zenodo [43]. A dataset containing recordings from all six sensor nodes has been published and is available on Dataverse [44]. To reproduce the results of this paper, the software may be used in combination with the dataset. As all measured values are available online in the log-files, only some key values will be listed in this section.

In Table 2, the measured distance and standard deviation are shown for different materials and distances. ${\bar{C}}_{Z}$ is the mean depth measured by sensor node four (N4). Some of the accumulated depth measurements were very close to the sensor due to noise. These flying pixels were considered outliers and including them would offset the mean calculation towards the sensor. To avoid such bias caused by flying pixels, depth measurements more than 50 cm from the median were excluded.

The unbiased standard deviation of the sample is calculated as
(1)$$\begin{array}{c} s=\sqrt{\frac{1}{n-1}\sum_{i=1}^{n}\left(C_{z\;i}-\bar{C_{z}}\right)}, \end{array}$$
where *n* is the number of depth images used according to Table 1. The three materials measured were the white region of the retroreflective markers (ID 40 and ID 1), the orange flat metal surface of the robot tracks and a white sheet of printer paper. As the different surfaces are close to each other, distortion should be similar. Observations that can be made from Table 2, is that the standard deviation was overall lower for the retroreflective sheeting. Compared to the metal surface, the marker was five times more accurate on a short-range measurement but gave approximately the same results at a long-range measurement. However, the marker was around five times more accurate than paper for both short- and long-range. Further, it has been observed that the retroreflective sheeting increases detectability on IR images considerably as shown in Figure 3. However, this has not been benchmarked.

It was observed that the initial pose estimation was often incorrect when the marker was placed far from the camera as the corners were detected with low precision. However, by using the method described in Section 4.3, it was possible to select the best poses. Nevertheless, it should be noted that there were scenarios where all the detected poses were actually wrong. In this case, the selected pose will also be wrong.

An example of the best and worst pose estimations for marker ID 13 is shown in Figure 7. Here, 100 images were captured wherein the marker was recognized 21 times. Of these 21 pose estimations, only 14 were correctly oriented. However, the described method was able to select the transformation nearest to the true pose verified by the obtained depth data.

Using quaternion averaging improves the overall pose estimate by averaging the best post estimate of the three markers. If one of the used orientations is wrong as a result of incorrect ambiguity selection, it is still possible to refine the calibration using the averaged pose as an initial guess.

Figure 8 shows an example of how the developed automatic calibration is robust against ambiguity in marker detection. The figure origins from calibration log C where as much as 400 color images were used. The wrongfully detected marker was ID 1 captured by sensor N1. Typically it has been observed that when the rate of wrongful poses for a marker is high, the overall detection rate of the same marker is low. Nevertheless, in this case, the detection rate was 100%, i.e., 400 wrongful pose estimations were made. Thus, the best of these estimations must also be wrong.

If only one marker was used to align the sensor cloud with the world-cloud, refinement using ICP may have failed as the orientation of the point clouds would be too different. However, as seen in Figure 8a, the red point cloud from sensor N1 was only tilted due to the failed pose. This illustrates that the averaging of three markers increased the robustness of the developed solution. The result after ICP refinement is shown in Figure 8b.

All the following results are extracted from log C unless stated otherwise. Table 3 summarizes the final calibrated sensor poses where the sensors, N, are numbered 1 to 6. Positions are given by the coordinates x,y, and *z*, while the orientations are shown in Euler angles RotZ, RotY, and RotX. The Kinect V2 reference frame has its origin in the image plane where the *x*-axis increased to the right, *y*-axis increased down, and *z*-axis increased in the direction of the camera projection. Therefore, when applied in the described order, RotZ yields the direction in the horizontal plane, and RotX notes the vertical orientation. Measured positions are shown in $X_{M}$, $Y_{M}$, and $Z_{M}$, but as described in Section 4.5, there were some deficiencies in the measurement method. Therefore, these values are included as indicative information. Thus, *e* is the indicative Euclidean distance error between the calibrated position and the measured position.

The distances between the sensors and the ArUco markers are given in Table 4. Here, we show that the farthest detection distance, between sensor 6 and ArUco ID 13, is 8.754
m. Further, the poorest detection was conducted by sensor 6 while detecting ArUco ID 40 where the marker was detected in only 7% of the images.

The sensor poses from Table 3 are labeled from N1 to N6 in Figure 9. In this figure, RGB point clouds from all six calibrated sensors have been accumulated for enhanced visibility. Further, the twelve validation markers are labeled from A to L. The distances between the sensors and these validation markers range from 4.07
m (N5 to L) to 9.45
m (N6 to G). World origin is located in the bottom left corner of the figure with the *x*-axis to the right and *y*-axis up. The three ArUco markers are labeled with their IDs 1, 13 and 40.

Figure 10 shows the same environment from different perspectives. Figure 10a,b have accumulated RGB point clouds while Figure 10c,d are single colored point clouds. Each sensor had a different color. Thus, sensors N1–N6 were colored in the order red, green, blue, magenta, yellow and cyan, respectively. The same coloring was used for all figures and plots where sensors are color coded. The coordinate system displayed on each ArUco marker in both Figure 9 and Figure 10 was based on the world map and was to be considered the ground truth for the marker positions.

The *x*- and *y*-position estimates of all evaluated validation markers are visualized in Figure 11. The perspective is the same as in Figure 9. Thus, one can conclude where the sensors, obstacles etc. are placed. Ground truth is drawn with black circles and all measurements seem to be quite accurate. As it is difficult to see calibration error at this scale, the reader is referred to Figure 12 for a more detailed view.

Figure 12 gives a detailed view of the measured validation marker pose errors w.r.t. ground truth of the individual marker. Figure 12a,b show all the positions colored according to sensor number while Figure 12c,d show the same points colored according to validation marker. Further, Figure 12a,c show the error in the xy-plane while Figure 12b,d show the error in the xz-plane. Thus, by looking at all four plots, it is possible to determine the 3D position of every point, what marker it was aimed at and what sensor that provided the data. The reader is referred to the provided log for accurate values.

By interpreting Figure 12, it is possible to see that there were some outliers. In Figure 12a one can see that sensor N3 had several points with an error of near 40 mm or more. By examining the five worst points of N3, one can see from Figure 12c that the corresponding validation markers are D, F, G, J, and L. Finding the location of these markers and the position of sensor N3 in Figure 9 reveals that all the markers were near the edge of the sensors field of view. Thus, a bit less accurate measurement may be expected, but it may also indicate that the sensor needs a better intrinsic calibration to compensate for distortion.

Another reason the measurements along the edges of the environment are outliers, may be that they are outside the region used for calibration, i.e., the area between the calibration markers. Looking at Figure 12c,d, one can see that the largest error for validation marker E and H were below 30 mm. As these markers were placed between the three calibration targets, they should be good. Thus, the calibration targets should be placed as far from each other as possible. Alternatively, the world map can be appended with additional reflective patches, such as the validation markers, for an additional ICP refinement where the search radius is small. Figure 13 illustrates the distribution of errors for each axis and a Euclidean distance, *d*, for the total error.

The logs are summarized in Table 5. Here, it is shown that the median error for all logs (including outliers) is around 25 mm. The mean error, calculated after removal of flying pixels, is around 30 mm. Execution time for calibration depends mainly on the number of accumulated clouds to refine using ICP. However, the parameters were subject to tuning to achieve desired performance.

## 6. Discussion

Our approach builds on previous work, [28], where extrinsic calibration between sensors was performed in three steps: ArUco transformation estimation, ROI-based ICP, and complete cloud ICP. In this paper, we overcome all the drawbacks from previous work and change the method substantially. The initial ArUco transformation estimation in the current method includes a novel algorithm that utilizes multiple markers and multiple images where detection is done on SD images in the depth camera reference frame. The previous method used a single HD image of a single marker. In this paper, a padded mask was used to create an ROI for ICP refinement. The exact ROI ensured that we employ the precise depth measurements of the retroreflective surfaces only. As large portions of an ArUco marker is black, the padding was added to utilize the reflective white border of the targets such that more points were obtained in the calculated point clouds. In [28], all steps were performed between pairs of sensors. In this paper, we registered each sensor to a world point cloud generated based on generated ArUco images and known poses. In doing so, it enables sensors to be calibrated in parallel. Further, this method removes the accumulation of errors and the need to consider angles between sensors facing the same target. The previous method did not include any calibration between sensors and the world, while the current method inherently registers to the world coordinate system. The previous method needed an additional step using the full point clouds for refinement, but when sensors are placed, e.g., facing each other, they will typically see different sides of an object. The different view makes the ICP refinement less precise than if all points were captured on surfaces observed by both sensors. The new method has removed this source of error as a world cloud is used for reference. Thus, the refinement can be completed directly on the ROI-based ICP step.

The results presented in Table 2 show that the retroreflective coating can reduce the standard deviation of depth measurements with a factor of five compared to other materials. Furthermore, we have also illustrated that the material has significantly higher visibility on IR images. By making planar fiducial markers using retroreflective coating, detectability, accuracy, and robustness are increased. We have not found a similar application of squared-based retroreflective fiducial markers and RGB-D cameras in the literature. The most common application of retroreflective markers we found in the literature was small spheres for motion capture systems such as [32]. The retroreflective planar boards presented in [18] use similar retroreflective coating as the proposed markers. However, the boards have no detectable ID, and a different sensor type was used. The most similar application we could find was the vertical precision landing concept presented in [33]. Here, the authors use an array of squared-based retroreflective fiducial markers. However, the markers are closely grouped to fit a landing platform while we want to separate the markers as much as possible within the view of the cameras. Furthermore, [33] use a monocular camera while we utilize RGB-D cameras.

The ambiguity problem is prominent when we detect markers at a distance such that the corner estimation of the marker is inaccurate. Further, the ArUco library is restrictive to avoid detecting false positives which in turn may yield fewer detections. Table 4 listed the relation between distance and detection rate in our experiments where the farthest distance was 8.75
m (N6, ID 13) and the lowest detection rate was 7% (N5, ID 40). Although a detection rate of 7% gives only 28 images from 400, we know that at least one of these had the correct orientation. This can be seen by studying Figure 8a closely where sensor N5 has a yellow color and ArUco marker ID 40 is in the background. In this figure, the farthest detection (N6, ID 13) can be observed to the left in cyan color. Sensor N1 detected the ambiguous wrongful pose of marker ID 1 on 400 of 400 correctly identified detections. This can be replicated using the published dataset. However, the application of pose averaging mitigated the wrongful pose.

The method to mitigate ambiguity in [29] seems to work well for short distances and can operate in near real-time as only one image is used. We have demonstrated superior results at more than twice the rated range of the sensor. However, we use a large set of images as our proposed application does not need to run in real-time. Further, we increase the robustness by averaging multiple markers while the method in [29] is currently made for a single marker. We have demonstrated the ability to solve the ambiguity problem using 60 cm markers at a range up to 8.75
m while [29] demonstrate zero errors up to 1.25
m using a 7 cm marker. It is possible that our method is more robust against small cross-calibration errors between RGB and depth as we do not use RGB for refinement, but this claim has not been validated.

The purpose of the presented method has been to increase autonomy and robustness of extrinsic calibration to a level where it may be applied automatically in an industrial setting. In the literature described in Section 2 there are several factors preventing system autonomy. Firstly, [4,7,16] use monocular cameras only. These methods are typically not accurate enough, and additional refinement using depth data should be added. Secondly, most of the methods require a human to be present in the environment to function as a key point or to move an object to be targeted [4,5,6,10,45]. Lastly, several of the methods are invasive in terms of cluttering the environment with installations such as spheres [7,8]. Permanently mounted planar markers are also invasive, but they are easier to place in a non-obstructive way. Such as in this paper, the robots are free to move along the tracks even if they are equipped with ArUco markers. The current setup typically uses around one minute per sensor and requires no human involvement except starting the program.

The accuracy of calibration systems using 3D sensors is often given as an average Euclidean distance between calculated and measured points. This error depends for example on the method, sensor technology, and the distance between the sensor and the measured points. As described in Section 2, Desai et al. [6] achieved an error of 1.5
cm at short range using RGB-D cameras. Tong and Barfoot [18] placed retroreflective landmarks 360${}^{\circ }$ around a 3D LiDAR (Light Detection And Ranging) sensor and got an error of 2 cm to 6cm. In this paper, we have demonstrated an average Euclidean error of 3 cm at distances up to 9.45
m using RGB-D sensors which is better than the described literature. The error between calibrated and measured sensor positions in Table 3 shows a position error of 25 mm to 93 mm. Although the measured positions are only indicative, we can conclude that the errors are realistic as a small position error can compensate for a possible small angle error in the calibration result.

## 7. Conclusions and Future Work

In this paper, we presented a fully automatic and robust calibration scheme for calibrating an arbitrary number of RGB-D sensors w.r.t. a world coordinate system. With autonomy in mind, we designed the system to be non-invasive in industrial operations. Thus, no personnel is required. An autonomous calibration enables the development of automatic re-calibration which is vital for industrial deployment.

We demonstrated the system by calibrating a sensor network comprised of six sensor nodes based on the active IR time-of-flight camera Kinect V2. These were positioned in a relatively large industrial robot cell with an approximate size of 10 m×10 m×4
m. The automatic calibration achieved an average Euclidean error of 3 cm at distances up to 9.45
m. We have not found any other non-invasive automatic calibration scheme in the literature that documents this level of accuracy using the Kinect V2 sensor.

Further, we have shown that using retroreflective sheeting enhances the accuracy of depth measurements based on active IR time-of-flight. Although this result may seem obvious, we have found no publications utilizing multiple retroreflective fiducial markers to increase calibration accuracy in the literature. Consequently, to the best of the authors’ knowledge, this paper presents a novel application for utilizing a grid of squared-based retroreflective fiducial markers.

We have demonstrated that the ambiguity problem can be mitigated by using the depth data to select the best available pose estimation from multiple RGB images. The solution is also robust against wrongful pose detection from a single marker as multiple markers are used.

The automatic calibration presented in this paper can easily be reproduced. The complete software code written using C++ for ROS, including experiment logs and instructions, is available open source [43]. Further, a replication dataset containing sensor data from all sensors in this paper is also available [44].

Future work to further improve the automatic calibration scheme can consist in decentralizing the calibration software such that it is run directly on the sensor nodes. Decentralization would increase the scalability of the system. Additional retroreflective landmarks may be added for an additional refinement to remove outliers due to extrapolation. For larger environments, the code should be expanded to work with any number of markers where it is only required that each sensor can see at least three markers for robust initial pose estimation. Redundant markers would increase the ability to do automatic re-calibration even if some markers are occluded or outside the sensor’s field of view. The software may also be further improved by including different sensor types such as LiDAR devices. Additional work can also be done on analyzing what the best method for overcoming the ambiguity problem using depth sensors is. For example, to evaluate if an initial pose estimate calculated using PnP directly on all marker key points is more accurate than calculating the individual marker poses and then fusing them using quaternion averaging. Further, the software should be tested in an outdoor environment such as depicted in Figure 2 to verify that the solution is robust enough to overcome sensor issues that may arise in, e.g., direct sunlight, shadows and other variations in weather conditions. For rough environments, where the sensors may move unexpectedly, there is also a need to implement an automatic warning or re-calibration.

## Figures and Tables

**Figure 1 sensors-19-01561-f001:**
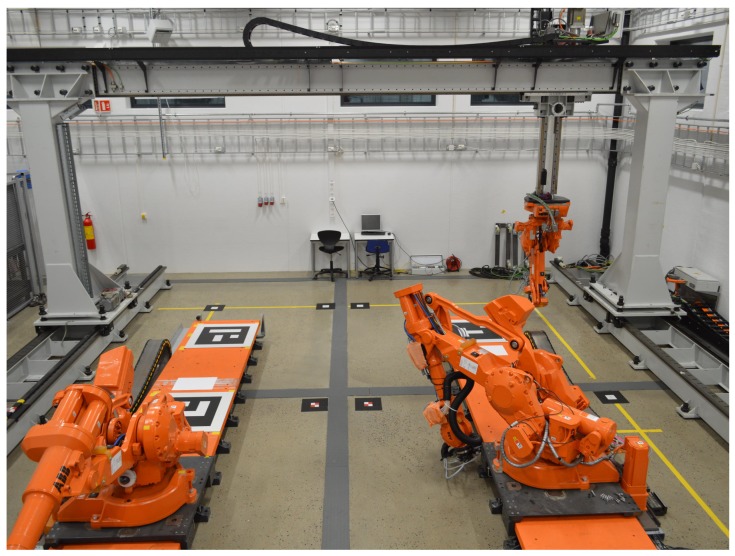
The Industrial Robotics Lab at the University of Agder. This lab consists of two rail-mounted ABB IRB4400 robots, one ABB IRB2400 robot mounted on a GÜDEL gantry and a processing facility.

**Figure 2 sensors-19-01561-f002:**
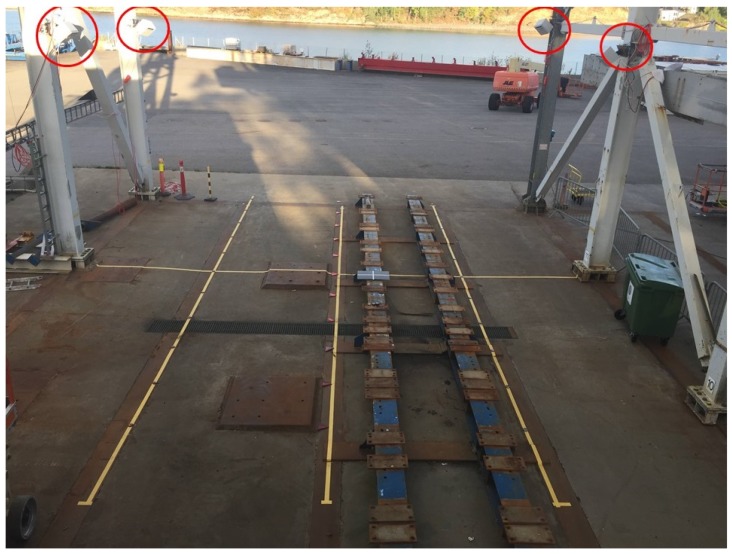
Example of an outdoor industrial environment. Four of the six sensor nodes are marked with red circles.

**Figure 3 sensors-19-01561-f003:**
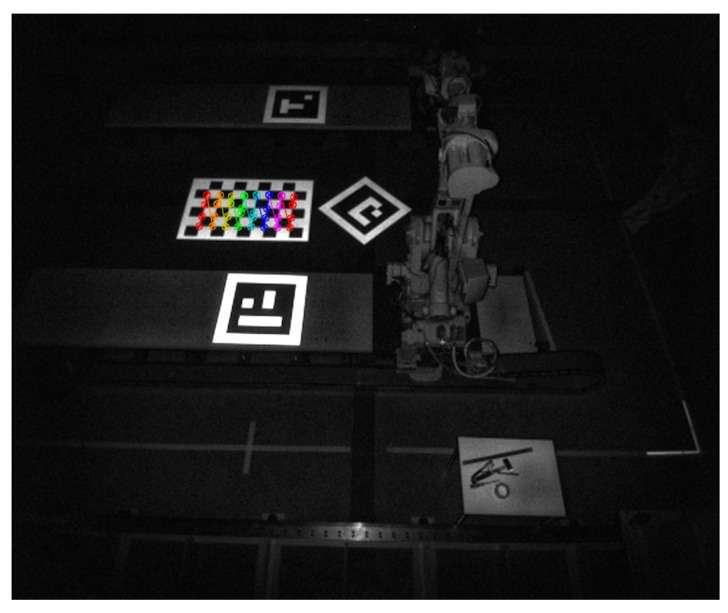
Detection of a chess pattern on retroreflective calibration targets in infrared (IR) image.

**Figure 4 sensors-19-01561-f004:**
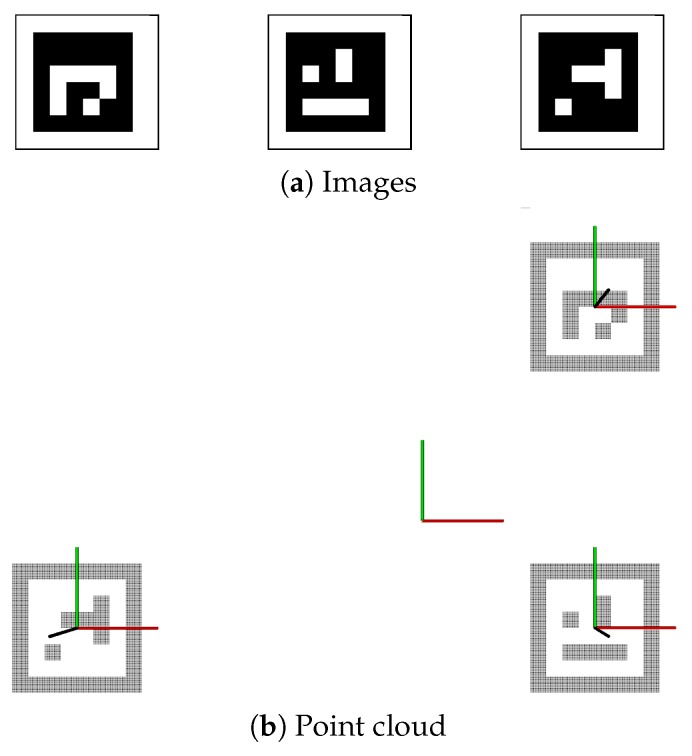
The world map was generated using the padded images of ArUco ID 1, 13 and 40 (**a**) where white pixels become points in the world reference cloud (**b**).

**Figure 5 sensors-19-01561-f005:**
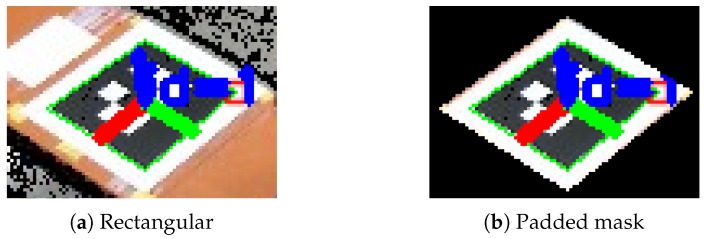
Rectangular crop compared to the padded mask crop.

**Figure 6 sensors-19-01561-f006:**
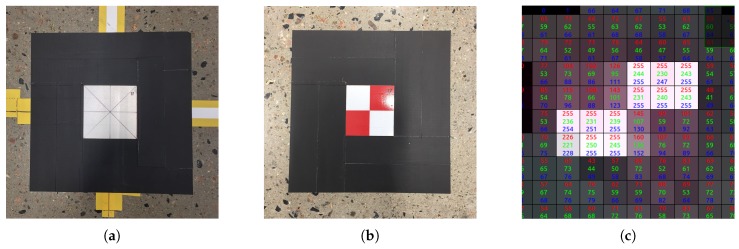
The retroreflective markers used for validating the calibration result. (**a**) White; (**b**) red and white; (**c**) red and white as seen from sensor N3.

**Figure 7 sensors-19-01561-f007:**
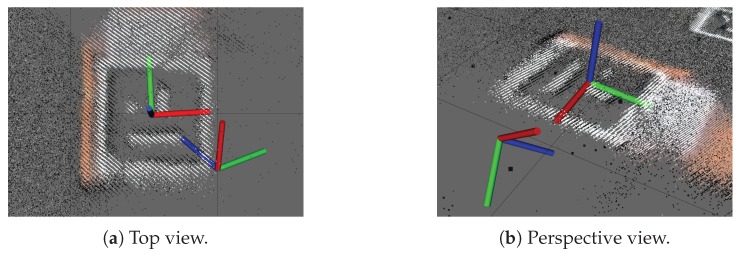
Pose ambiguity where the best and worst pose estimation from 21 detections are shown.

**Figure 8 sensors-19-01561-f008:**
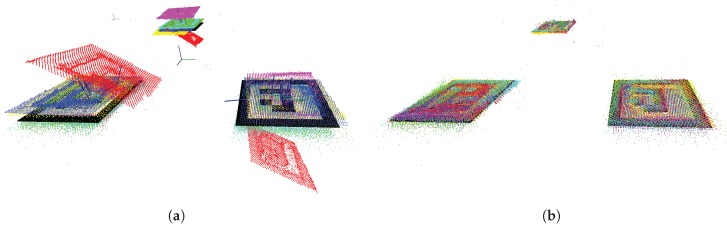
Two calibration steps for all six sensor nodes registers the sensor point clouds to the world map (black). The data reflects calibration log C. (**a**) ArUco-based calibration with one wrongful pose; (**b**) iterative closest point (ICP) refined calibration.

**Figure 9 sensors-19-01561-f009:**
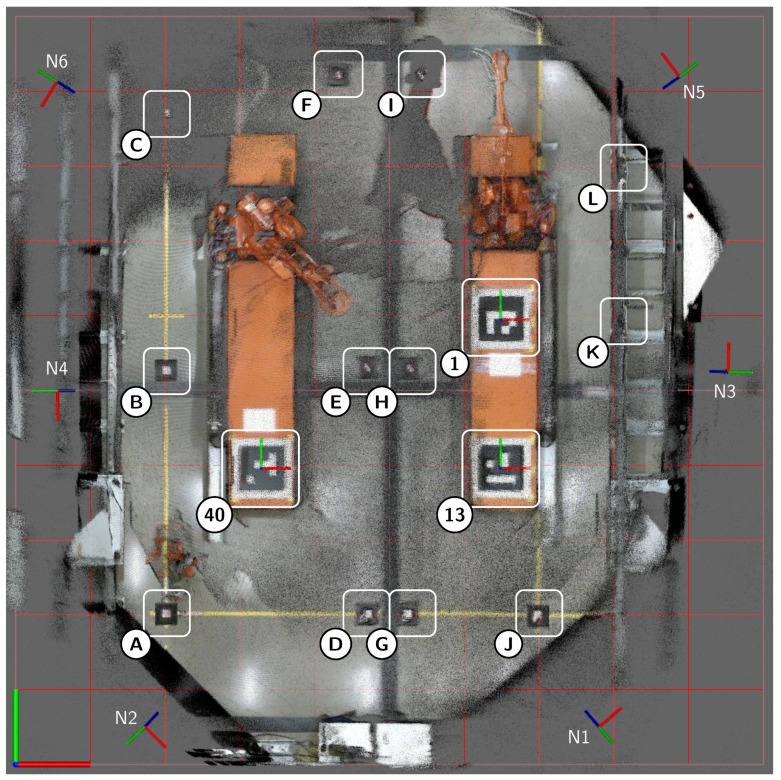
Orthogonal view of the combined accumulated RGB point clouds of all six sensors after calibration. Sensor node poses are annotated N1-N6 and validation targets A-L. ArUco markers are labelled according to ArUco ID.

**Figure 10 sensors-19-01561-f010:**
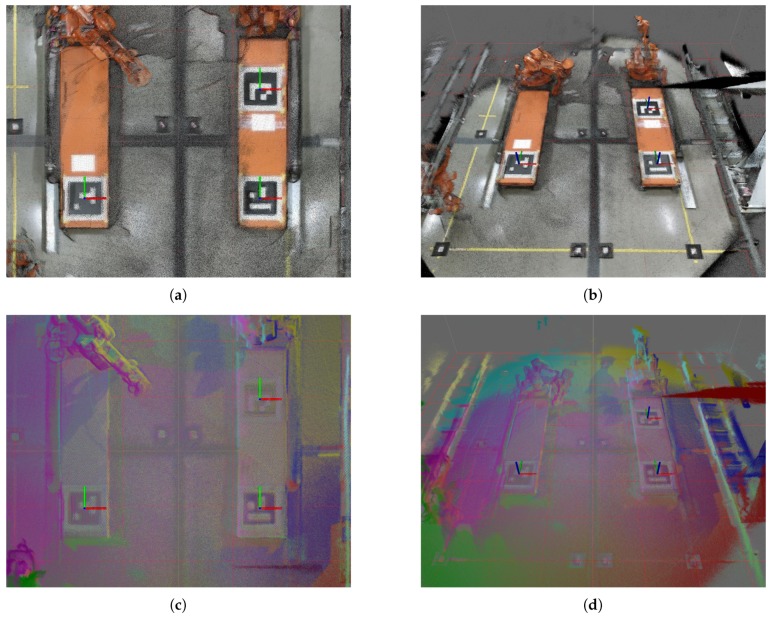
Accumulated point clouds of all six sensors after calibration. (**a**) RGB point clouds, orthogonal view; (**b**) RGB point clouds, perspective view; (**c**) Single color point clouds, orthogonal view; (**d**) Single color point clouds, perspective view.

**Figure 11 sensors-19-01561-f011:**
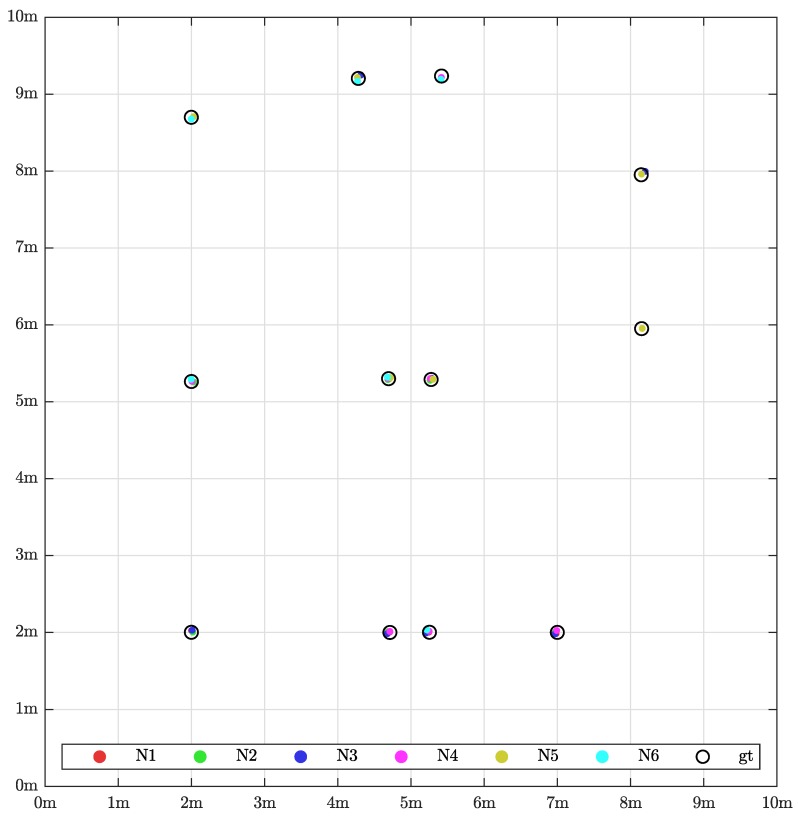
Orthogonal projection of the xy-plane showing the detected sensor validation points against the ground truth measured with a Leica AT960 laser tracker. See Figure 9 for illustration of sensor placement. For interpretation of the references to color in this figure legend, the reader is referred to the web version of this paper.

**Figure 12 sensors-19-01561-f012:**
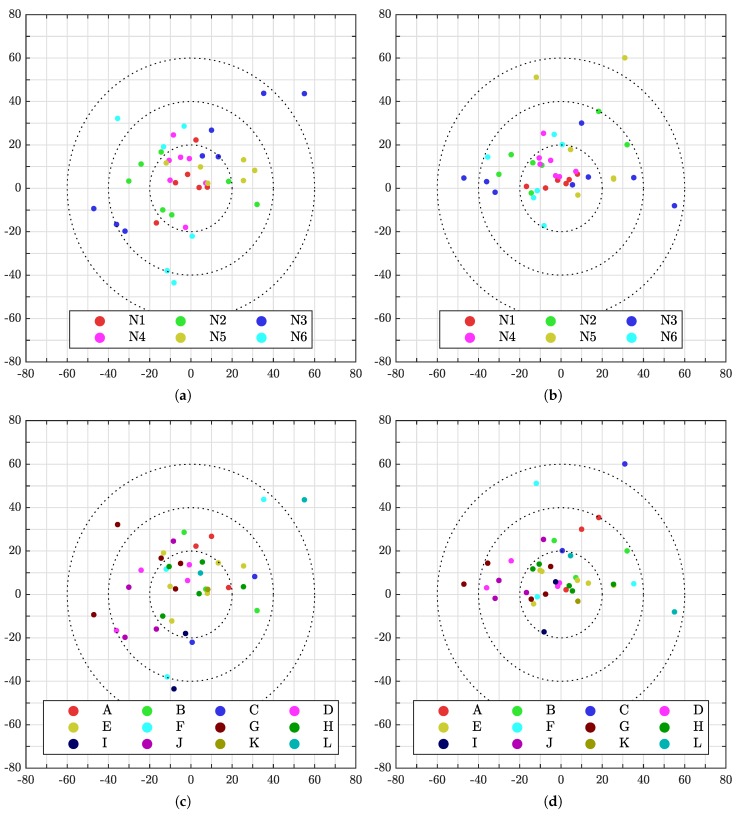
Errors of all measured validation points. For interpretation of the references to color in this figure legend, the reader is referred to the web version of this paper. (**a**) Errors in mm per sensor in the xy-plane; (**b**) errors in mm per sensor in the xz-plane; (**c**) errors in mm per validation marker in the xy-plane; (**d**) errors in mm per validation marker in the xz-plane.

**Figure 13 sensors-19-01561-f013:**
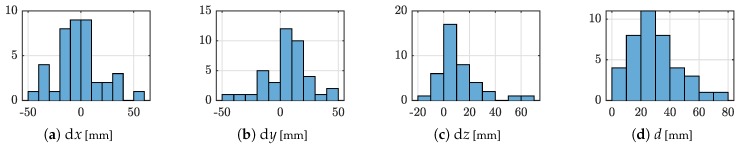
Error distribution of the calculated calibration error w.r.t. world axes x,y,z and the Euclidean distance d. Histogram *x*-axis is divided in 10 mm bins and *y*-axis is the number of occurences.

**Table 1 sensors-19-01561-t001:** Parameters used to record three log files.

Log	A	B	C
ArUco marker size:	0.6 m	0.6 m	0.6 m
Color images per node	400	400	400
Depth images per node	50	100	100
Point clouds per node	10	20	10
ICP, starting search radius, $ICP_{D}$	1.0 m	1.0 m	1.0 m
ICP, maximum rounds, $ICP_{R}$	10	10	10
ICP, maximum iterations in first round, $ICP_{IT}$	100	100	100
ICP, convergence limit, $ICP_{CC}$	10^−8^	10^−12^	10^−8^
ICP, transformation step size, $ICP_{TE}$	10^−14^	10^−18^	10^−14^

**Table 2 sensors-19-01561-t002:** Example of standard deviation for different materials and distances.

Log	A		B		C	
	${\bar{C}}_{Z}$ [m]	s [mm]	${\bar{C}}_{Z}$ [m]	s [mm]	${\bar{C}}_{Z}$ [m]	s [mm]
Retroreflective	4.803	1.4	4.796	1.4	4.796	1.3
Retroreflective	6.690	6.4	6.690	5.1	6.695	5.0
Metal	4.814	8.2	4.808	6.2	4.810	6.7
Metal	6.690	6.4	6.690	5.1	6.695	5.0
Printer paper	4.809	7.1	4.802	7.5	4.802	7.4
Printer paper	6.731	29.3	6.719	22.7	6.717	21.0

**Table 3 sensors-19-01561-t003:** Automatically calibrated sensor positions in meter and orientations in degrees.

N	X	Y	Z	RotZ	RotY	RotX	X${}_{M}$	Y${}_{M}$	Z${}_{M}$	*e* [mm]
1	7.798	0.496	4.175	40.170	0.639	−136.170	7.841	0.561	4.180	78
2	1.729	0.501	4.135	−45.253	2.063	−141.490	1.710	0.470	4.158	43
3	9.522	5.275	4.353	88.439	−0.072	−143.461	9.484	5.215	4.293	93
4	0.553	4.995	4.353	−89.208	−0.495	−145.591	0.560	4.998	4.377	25
5	8.879	9.192	4.194	126.733	−1.311	−139.272	8.900	9.181	4.157	44
6	0.559	9.136	4.145	−121.458	−0.487	−137.562	0.540	9.160	4.176	44

**Table 4 sensors-19-01561-t004:** Euclidean distance from calibrated sensor positions to known positions of ArUco markers and their respective detection rates.

	ID 1		ID 13		ID 40	
N	d [m]	Rate [%]	d [m]	Rate [%]	d [m]	Rate [%]
1	6.809	100.0	5.353	100.0	6.878	99.75
2	8.179	72.5	7.015	99.75	5.385	100.0
3	5.101	100.0	5.225	100.0	7.550	100.0
4	7.241	100.0	7.253	100.0	4.987	100.0
5	5.600	100.0	6.943	100.0	8.590	7.0
6	7.746	94.5	8.754	40.0	6.996	100.0

**Table 5 sensors-19-01561-t005:** Overall comparison between the three log files.

Log	A	B	C
Time to record data	20.4 s	30.2 s	28.9 s
Time to process images	10.2 s	10.2 s	8.9 s
Time to calculate ArUco transformations	2.2 s	3.6 s	2.2 s
Time to perform ICP refinement	291.7 s	1411.5 s	649.7 s
Median value of error distances	25.5 mm	23.9 mm	24.9 mm
Mean value of error distances	29.7 mm	30.4 mm	29.6 mm
Standard deviation of error distance set	19.0 mm	17.9 mm	16.3 mm

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
