# Peer review of "Automatic Calibration of an Industrial RGB-D Camera Network Using Retroreflective Fiducial Markers"

_sensors, 2019, doi:10.3390/s19071561_

Round 1
Reviewer 1 Report
Main novel contributions of the paper seem to be published in [5] which is the authors' own recent contribution. However, it is hard to find a copy of the reference but judging from the presenting of the paper at hand, most significant contributions of the paper are already published. The paper claims to contribute to the already published results in three trivial ways for example by considering a smaller sensor coverage comparing to what has been published. The paper reads like a developer’s manual as oppose to a presentation which can be included in a scientific journal. The technical presentation of the paper is rather low, and it basically focuses on somewhat of an ad-hoc experimentation of the existing methods and tools (e.g. OpenCV, ROS, ..). For example, the paper mentions large angle between the cameras without any specifics on their magnitudes. It is not clear why this step was carried: First, a world map is created by converting padded ArUco marker images to point clouds and transforming them to a known position and orientation.
This statement is not clear and is not supported by any technical discussions: Three planar fiducial markers were made so that the planar ROI point cloud would be the same regardless of sensor pose.
It is discussed in the literature that images of sphere (such as its point cloud) will have similar correspondences between camera frames. However, it is not clear why the paper claims this statement without any proves: Non-planar markers, such as a sphere, would yield different point clouds depending on the sensor pose.
The error characterization of the algorithm is not clear. For example, Figure 13 shows errors are in mm in the maximum error range of 60 mm. however, in the discussion section, it is concluded that the mean calibration error was found to be around 3m.
Overall, the paper presents some experimental results of an already exiting algorithm without any comparison with other similar calibration methods presented in the literature review (and with method of spheres).
Author Response
Please see attached PDF document.
Best Regards,
Atle Aalerud,
Corresponding Author.

Reviewer 2 Report
1-Introduction section
Introduction should be revised, and several references related to the topic should be cited
such as:
(1) S. Foix, G. Alenya and C. Torras, "Lock-in Time-of-Flight (ToF) Cameras: A Survey," in IEEE Sensors Journal, vol. 11, no. 9, pp. 1917-1926, Sept. 2011.
doi: 10.1109/JSEN.2010.2101060
(2) Jung, J., Yoon, I., & Paik, J. (2016). Object Occlusion Detection Using Automatic Camera Calibration for a Wide-Area Video Surveillance System. Sensors (Basel, Switzerland), 16(7), 982. doi:10.3390/s16070982
(3) Guan, J., Deboeverie, F., Slembrouck, M., Van Haerenborgh, D., Van Cauwelaert, D., Veelaert, P., & Philips, W. (2016). Extrinsic Calibration of Camera Networks Based on Pedestrians. Sensors (Basel, Switzerland), 16(5), 654. doi:10.3390/s16050654
(4) Tan, L., Wang, Y., Yu, H., & Zhu, J. (2017). Automatic Camera Calibration Using Active Displays of a Virtual Pattern. Sensors (Basel, Switzerland), 17(4), 685. doi:10.3390/s17040685
(5) Bi, S., Yang, D., & Cai, Y. (2018). Automatic Calibration of Odometry and Robot Extrinsic Parameters Using Multi-Composite-Targets for a Differential-Drive Robot with a Camera. Sensors (Basel, Switzerland), 18(9), 3097. doi:10.3390/s18093097
3- Calibration method section
-This section is not written well and should be revised because it contains many of spelling corrections which are needed such as in lines:
226 buildt
328 se
334 the the
335 Looking at Figs. 13c and 13c,
354 Cropping the the depth
355 world map map
5- discussions and conclusions
The reader does not really understand where the discussion is; and the conclusions are not clear; when the reader reads only section 5, it is not obvious to understand what has been done in the field. The sentence about accuracy is misleading (2 m ?) Accuracy refers to an error; how is the error defined.
-Also the proposed method is lacking to some mathematical notations (such as equations) describe the proposed method to be clear for reproducibility by simple readers and researchers in robotics field.
-Figures 7,8,9 need to rearranged because fig 9 is mentioned before fig 7 and fig 8.
-Datasets A, B and C are collected in indoor industrial robots environment only which is not adequate to judge on the proposed method is applicable directly in outdoor industrial robots environment as described in Fig. 2 which may contain different natural lighting conditions that are not considered in the experiments such as direct sunlight, shadows and other variations in weather such as mist.
Author Response

(The authors gave the same response as above.)

Reviewer 3 Report
Overall impression of the paper is very good. With little correction it could be improved.
1. In the introduction I am missing strong motivation way to develop such an approach and what target accuracies should be achieved (i.e. +/- 1mm at volume 10x10x4m).
2. Why ArUco visual markers are placed on flat surfaces, but not on the robot arms heads (as it is seen from Figure 1). System calibration with robot arm could give better precision, because robot-to-world coordinates accuracy is less than 25px.
3. Still missing information on the ArUco markers - whay they are flat? If you would use the cube with 3 markers, it could increase marker visibility and recognition. What was the requested accuracy for markers detection?
4. In the text line 256 first time ever is used unexplained notation “N4” and also please explain effect of “flaying pixels”.
5. Table 3 presents calibrated sensors positions; Can they be compared with real world measurements and what level of difference is between real and calibrated measurements?
6. Fig. 12 -13, what was the target accuracy?
7. In text lines 358-359 the mean calibration accuracy is 3-7m? So bad?
8. Finally, missing guidance for future work or suggestion for the improvements.
Author Response

(The authors gave the same response as above.)

Round 2
Reviewer 1 Report
authors have added some explanations and extended their presentation of the results.
Overall the main contributions of the paper have been published by the authors before, but, this version can be considered as an extended version of their previous publication.